# Lineage tracing of genome-edited alleles reveals high fidelity axolotl limb regeneration

Grant Parker Flowers[1]*, Lucas D Sanor[1], Craig M Crews[1,2,3]*

[1]Department of Molecular, Cellular and Developmental Biology, Yale University, New Haven, United States; [2]Department of Chemistry, Yale University, New Haven, United States; [3]Department of Pharmacology, Yale University, New Haven, United States

**Abstract** Salamanders are unparalleled among tetrapods in their ability to regenerate many structures, including entire limbs, and the study of this ability may provide insights into human regenerative therapies. The complex structure of the limb poses challenges to the investigation of the cellular and molecular basis of its regeneration. Using CRISPR/Cas, we genetically labelled unique cell lineages within the developing axolotl embryo and tracked the frequency of each lineage within amputated and fully regenerated limbs. This allowed us, for the first time, to assess the contributions of multiple low frequency cell lineages to the regenerating limb at once. Our comparisons reveal that regenerated limbs are high fidelity replicas of the originals even after repeated amputations.

DOI: https://doi.org/10.7554/eLife.25726.001

**\*For correspondence:**
grant.flowers@yale.edu (GPF);
craig.crews@yale.edu (CMC)

**Competing interests:** The authors declare that no competing interests exist.

## Introduction

The origin of the cells of the blastema, the mass of proliferating cells that gives rise to regenerated structures, has long been the subject of debate (*Tanaka and Reddien, 2011*). In salamanders, such as the axolotl, this debate has centered on whether the limb blastema arises from dedifferentiated cells or proliferating stem cells (*Tanaka, 2016*). Earlier studies using tissue grafts suggested that restricted tissue types, such as dermis or cartilage, can give rise to some or all of the other tissues in regenerated limbs (*Namenwirth, 1974*; *Muneoka et al., 1986*). More recent work using GFP-labelled grafts found little evidence of lineage switching during limb regeneration, though dermis may contribute to cartilage and tendons (*Kragl et al., 2009*). A study using tissue grafting and in vivo imaging of single cells during digit tip regeneration further suggests that dedifferentiated or proliferative chondrocytes contribute very little to regeneration, and these cells are most likely replenished by dermal fibroblasts (*Currie et al., 2016*). Comparisons of muscle regeneration between species of salamanders have shown that axolotls and larval newts regenerate muscle by recruiting satellite cells to the blastema; however, fragmentation and cell cycle reentry of myofibers provides the cellular source of regenerated muscle in adult newts (*Sandoval-Guzmán et al., 2014*; *Tanaka et al., 2016*). This long history of grafting studies of salamander limb regeneration demonstrates that this process is governed by numerous tissue- and life-cycle-dependent mechanisms.

Here, we used a novel approach to simultaneously assess the contributions of many cell lineages to the entire regenerated limb. By quantifying embryonically-produced CRISPR/Cas-generated lesions in individual axolotl limbs before and after full regeneration, we determined the extent to which regenerated limbs are reconstituted by the same cell lineages that comprised the original limb. We find that a regenerated axolotl limb is a high fidelity replicate of the original limb comprised of the same cell lineages, with lineages that contribute to as little as one ten-thousandth of

the original limb found in the regenerated limb at the same relative frequencies. This level of fidelity is maintained after multiple rounds of amputation and full limb regeneration.

These observations, and our quantification of this fidelity, may be of great importance to future studies as the salamander is now genetically tractable, in particular by using CRISPR/Cas-mediated mutagenesis (*Flowers et al., 2014*). Numerous studies using high-throughput gene expression analysis identified an expanding set of genes specifically upregulated in the axolotl limb blastema (*Bryant et al., 2017*; *Knapp et al., 2013*; *Voss et al., 2015*). The abundance of uncharacterized targets of interest has created a demand for higher throughput methods to characterize the consequences of genetic manipulation on limb regeneration in mosaic F0 animals. This work establishes the baseline fidelity with which a regenerated axolotl limb is reconstructed with respect to embryonic lineages in such mosaic animals. In the future, a similar approach may be used to identify genes that when altered affect the ability of embryonic cell lineages to contribute to the regenerated limb.

## Results and discussion

Here, we used the CRISPR/Cas system to produce unique insertions and deletions at targeted genomic sites to independently genetically label numerous cell lineages in the developing axolotl embryo. We induced targeted mutations during early embryonic development by microinjecting *cas9* mRNA into the axolotl zygote with a single guideRNA (gRNA) against one of several genomic loci (*Figure 1a*). The start and end points of induced insertions and deletions (indels) frequently vary with each event, as do the precise identities of inserted sequences. Therefore, each indel labels the genome of the affected cell and its descendants (*Figure 1a*).

Using gRNAs effective at inducing high levels of mutagenesis that were not expected to have a deleterious impact on limb regeneration, we targeted two to four animals with one of six gRNAs to produce 19 animals with targeted mutations. We extracted DNA from amputated original and fully regenerated limbs from the same individuals and performed Next Generation Sequencing (NGS) to identify all indels generated at the targeted sites for each gRNA. NGS of PCR products of targeted genomic loci reveals that DNA extracted from the entire amputated limbs of mutagenized axolotls is mosaic and exhibits allelic heterogeneity (*Figure 1a,b*; *Figure 1—figure supplement 1*; mean = 27.3 alleles, s.d. = ±10.2, n = 19 original limbs). A simple linear regression indicates that the log score of the normalized read numbers for each allele in the primary limb predicts the log score of the normalized read number of each allele in the secondary limb ($r^2$ = 0.37, p<1 $\times$ 10$^{-15}$). Thus, specific-allele-labelled cell populations contribute to both the original and regenerated limbs at similar frequencies. The majority of the alleles detected in the limbs of animals in this study are low frequency (i.e., represent less than 1.6% of all reads in the limb; 367/492 alleles, *Figure 2a*), with a median of 2332 reads per million (RPM). When we examined the behavior of these alleles, we found that low frequency alleles are more likely to increase in frequency or exhibit less than a one fold decrease (197/367 alleles) than to disappear (97/367 alleles; *Figure 2b*). Taken together, our work indicates that low frequency cell lineages are highly preserved in the limb after regeneration.

This approach allows us to quantify the fidelity of limb regeneration with respect to cell lineages. *Figure 3a* depicts the relative frequencies of sequence reads corresponding to individual mutant and non-mutant alleles found in the limb of a single animal before (blue-filled circle) and after (red open circle) complete limb regeneration. The relative contribution of each allele to the limb remains remarkably constant after full regeneration. In this representative case, the net difference in the RPM for all alleles before and after complete limb regeneration represents 10.1% of the total reads (*Figure 3b*). We found that among the 19 animals in this study, the mean difference in allelic identities of original and secondary limbs was only 24.2% (s.d. = ±16.1%; *Figure 4c*). Interestingly, the original limb of one animal, *EGFP* 4, exhibited spontaneous malformations and polydactyly at the time of the first amputation (*Figure 4c*, *Figure 4—figure supplement 1*) and showed the highest change in allelic identity between limbs of all assessed animals, 63.3%. This suggests that the change in allelic identity reflects the morphological fidelity of regeneration. Removing this case reduces the mean difference in allelic identity between original and secondary limbs to 22.0% (s.d. = ±13.6%). There is no significant correlation between overall mutation rate or number of alleles and the change in allelic identity of regenerated limbs ($R^2$ = 0.14, p=0.11 for mutation rate and identity; $R^2$ = 0.01, p=0.76 for mutation rate and total number of alleles). These findings demonstrate that a newly regenerated limb is a surprisingly high fidelity replica of the original limb.

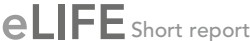

**Figure 1.** Comparison of CRISPR-generated alleles in original and regenerated axolotl limbs. (**A**) Schematic of alleles generated by CRISPR-induced mutagenesis in original and regenerated axolotl limbs. Axolotls were zygotically injected with *cas9* mRNA and single gRNA (top left), producing differing indels at the targeted site that are inherited in different cell lineages (top right, yellow depicts the putative cut site in an intact sequence of *egfp,* dashes depicted deleted sequence, red text depicts inserted sequences). This allelic mosaicism is revealed in NGS of an amputated limb. We allowed amputated limbs to regenerate to determine the extent to which the allelic mosaicism resembles that of the original limb (bottom). (**B**) Comparison of all alleles tracked in this study between original and regenerated limbs of 19 animals. The log scores of reads per million (RPM) of every allele in the original limb are significantly correlated with those of the secondary limb. All source data is found in *Figure 1—source data 1*.

DOI: https://doi.org/10.7554/eLife.25726.002

The following source data and figure supplements are available for figure 1:

**Source data 1.** All allele counts in all animals.
DOI: https://doi.org/10.7554/eLife.25726.008

**Figure supplement 1.** The log scores of reads per million (RPM) of every allele in the original limb compared with those of the secondary limb for each individual animal used in this study.
DOI: https://doi.org/10.7554/eLife.25726.003

**Figure supplement 2.** Characterization of the frequency with which alleles co-occur in different animals.
DOI: https://doi.org/10.7554/eLife.25726.004

**Figure supplement 2—source data 1.** Counts of alleles that are unique and shared between animals.
DOI: https://doi.org/10.7554/eLife.25726.005

*Figure 1 continued on next page*

*Figure 1 continued*

**Figure supplement 3.** Distribution of unique and shared indel sequences across animals injected with reduced concentrations of *cas9* and gRNAs.
DOI: https://doi.org/10.7554/eLife.25726.006
**Figure supplement 3—source data 1.** Counts of all alleles in animals injected with varying concentrations of cas9 and gRNAs.
DOI: https://doi.org/10.7554/eLife.25726.007

Salamanders such as axolotls and newts can regenerate the same structures repeatedly throughout life, though the rate and fidelity of such regeneration can decline with age or after metamorphosis (*Monaghan et al., 2014*; *Seifert and Voss, 2013*). Here, we addressed whether repeated limb amputations produce a decline in the fidelity with which a twice-regenerated limb replicates the original limb. After full regeneration of secondary limbs, we carried out a second amputation of the limbs of 12 embryonically mutagenized animals, permitted full tertiary limb growth, subsequently amputated those limbs, and used NGS to identify and quantify each allele at the targeted sites (*Figure 4a*). A simple linear regression reveals that the log score of the normalized read numbers for each allele in the primary predicts the log score of the normalized read number of each allele in the tertiary limb ($r^2$ = 0.47, p<$1 \times 10^{-15}$; *Figure 4b*). Furthermore, we found that the allelic identities of these tertiary limbs differed very little from those of the secondary limbs (mean = 15.8%, s.d. = ±6.5%; *Figure 4c*; *Figure 4—figure supplement 2*). Thus, just as there is not a decline in morphological fidelity after repeated amputations, there is no decline in the fidelity of cell lineages that comprise the limb.

We sought to investigate whether the alleles generated in this process label distinct or diffuse cell populations. Previous studies have indicated that CRISPR-induced indels are not truly random, as particular indels arise more frequently than others after mutagenesis with a single gRNA (*Gagnon et al., 2014*; *Burger et al., 2016*). However, comparing the identity of alleles across individuals shows that the majority of allele sequences appeared in only one animal out of the as many as four animals mutagenized with a single gRNA (*Figure 1—figure supplement 2a*). For gRNAs assessed in four animals, the individual alleles that received fewer than 1.6% of all sequence reads in an individual (which we will refer to as low frequency alleles) appeared in only one or two animals, while all alleles that received more reads per individual were found in multiple animals (*Figure 1—figure supplement 2b*). To further explore the allelic diversity produced at one site, we injected embryos with reduced quantities of pooled gRNAs and *cas9* and carried out NGS in mutant animals. At concentrations of *cas9* that produced no or single-digit numbers of alleles at each target in each embryo (mean = 1.4 alleles, s.d. =±1.6, n = 50, 5 targets tested in 10 animals each), the majority of alleles were unique (26/39 alleles; *Figure 1—figure supplement 3*). Together, these results suggest that, though some alleles are favorably produced from targeted mutagenesis, most alleles are formed with sufficiently low probabilities as to represent unique cell lineages even in highly mosaic mutant animals.

To address whether unique cell lineages are tissue-restricted, we carried out subsequent amputations from regenerated forelimbs of four mutagenized animals. After these amputations, from each animal we extracted blood samples, collected the hand from the amputated forelimb, and dissected nerve, blood vessel, bone/cartilage, skin, and muscle from the remaining amputated forelimb. We then extracted DNA from the hand and the tissue-enriched samples and carried out NGS of PCR products of the targeted genomic loci in each tissue. NGS revealed that the high frequency alleles (i.e., those representing greater than 6.5% of all reads) of the whole hand could be detected in all or almost all of the six individual tissue-enriched DNA samples from the same limb (mean = 5.4 of 6 tissues, s.d. = ±1.1, n = 15/171 alleles, *Figure 5—figure supplement 1a*). Conversely, alleles that appear at a low frequency within the hand (i.e., those that represent less than 1.6% of all reads) are found in fewer tissues (mean = 2.8 of 6 tissues, s.d. =±1.6, n = 135 alleles, *Figure 5—figure supplement 1a*). We find that each tissue-enriched sample contains alleles exhibiting more than a 16-fold enrichment in normalized reads relative to all other tissues (29/171 alleles), and 29 additional alleles were found to be similarly enriched in only two tissues. From all tissue samples, we found that the blood was the source of the most tissue-enriched alleles (17/58 alleles) likely arising from the distant origins of hematopoietic cells and the ease with which blood may be isolated (*Lopez et al., 2014*; *Figure 5—figure supplement 1b*)

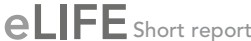

**Figure 2.** The sizes and fold changes of all alleles found in original and regenerated limbs in this study. (**A**) Pie chart depicting numbers of all alleles identified in original limbs binned by RPM. The majority of alleles detected are low frequency (i.e., represent less than 1.6% of all sequence reads in limb). (**B**) Histograms depicting the log of fold change for all alleles after limb regeneration. The majority of low frequency alleles are retained.
DOI: https://doi.org/10.7554/eLife.25726.009

**A**

Relative allele frequency in original limb

Relative allele frequency in regenerated limb

```
ACTGGTTCAACCTGGAGAAGTTCCTGAAGGAGGCGGACCGCATCCTCAAACCCAATGGCTGTCTTGC
ACTGGTTCAACCTGGAGAAG------------GCGGACCGCATCCTCAAACCCAATGGCTGTCTTGC
ACTGGTTCAACCTGGAGAAGT-------------GGACCGCATCCTCAAACCCAATGGCTGTCTTGC
ACTGGTTCAACCTGGAGAAGTT-------------GACCGCATCCTCAAACCCAATGGCTGTCTTGC
ACTGGTTCAACCTGGAGAAGTTCC----------GGACCGCATCCTCAAACCCAATGGCTGTCTTGC
ACTGGTTCAACCTGGAGAAGTTCC--------------CGCATCCTCAAACCCAATGGCTGTCTTGC
ACTGGTTCAACCTGGAGAAGTTCCT-------------------------------------TGC
ACTGGTTCAACCTGGAGAAGTTCCTGG--------------------------------------
ACTGGTTCAACCTGGAGAAGTTCCTGA----GGCGGACCGCATCCTCAAACCCAATGGCTGTCTTGC
ACTGGTTCAACCTGGAGAAGTTCCTGACC---GCGGACCGCATCCTCAAACCCAATGGCTGTCTTGC
ACTGGTTCAACCTGGAGAAGTTCCTGA------CGGACCGCATCCTCAAACCCAATGGCTGTCTTGC
ACTGGTTCAACCTGGAGAAGTTCCTGACC----------GCATCCTCAAACCCAATGGCTGTCTTGC
ACTGGTTCAACCTGGAGAAGTTCCTGAACCTGAAAGGGATGGGAGGAGCCAGCCCTCCACATGGC
                                                             TGTCTTGC
ACTGGTTCAACCTGGAGAAGTTCCTGAAC---GCGGACCGCATCCTCAAACCCAATGGCTGTCTTGC
ACTGGTTCAACCTGGAGAAGTTCCTGAA--------ACCGCATCCTCAAACCCAATGGCTGTCTTGC
ACTGGTTCAACCTGGAGAAGTTCCTGAA---------CCGCATCCTCAAACCCAATGGCTGTCTTGC
ACTGGTTCAACCTGGAGAAGTTCCTGAAACCA-------CATCCTCAAACCCAATGGCTGTCTTGC
ACTGGTTCAACCTGGAGAAGTTCCTGAAGCCATTGGGTGT
                              GAGGCGGACCGCATCCTCAAACCCAATGGCTGTCTTGC
ACTGGTTCAACCTGGAGAAGTTCCTGAAG-AGGCGGACCGCATCCTCAAACCCAATGGCTGTCTTGC
ACTGGTTCAACCTGGAGAAGTTCCTGAAGTT----GACCGCATCCTCAAACCCAATGGCTGTCTTGC
ACTGGTTCAACCTGGAGAAGTTCCTGAAGG---CGGACCGCATCCTCAAACCCAATGGCTGTCTTGC
ACTGGTTCAACCTGGAGAAGTTCCTGAAGGT---GGACCGCATCCTCAAACCCAATGGCTGTCTTGC
ACTGGTTCAACCTGGAGAAGTTCCTGAAGGA-----ACCGCATCCTCAAACCCAATGGCTGTCTTGC
ACTGGTTCAACCTGGAGAAGTTCCTGAAGGA------CCGCATCCTCAAACCCAATGGCTGTCTTGC
```

**B**

Cumulative change in allelic identity = 10.1%

**Figure 3.** Comparison of all alleles in primary and secondary limb in one representative animal. (**A**) Depiction of all identified alleles and their relative sizes in primary and secondary limbs of one individual mutagenized with *meth t2* gRNA. The sequence of the gRNA and protospacer adapter motif of the intact highlighted sequence are depicted in yellow. The relative number of reads of each allele in the original (filled blue circle) or secondary limbs

*Figure 3 continued on next page*

*Figure 3 continued*

(open red circle) are depicted to the right of each allele. (B) The cumulative difference between all alleles of original and secondary limbs for the depicted animal represents 10.1% of all reads (filled black circle, drawn to the same scale employed in A).

DOI: https://doi.org/10.7554/eLife.25726.010

We conducted Fisher's exact tests to determine whether such enriched alleles found in particular tissues were disproportionately represented among alleles that either increased or decreased in frequency after limb regeneration. We found that the only source of tissue-enriched alleles that underwent any significant change was skin, as skin-enriched alleles were disproportionally represented both among alleles that decreased in frequency from primary to secondary limbs and among those that increased in frequency from secondary to tertiary limbs (skin-enriched alleles = 14/58 tissue-enriched alleles, 8/14 tissue-enriched alleles decreasing from primary to secondary, 9/17 alleles increasing from secondary to tertiary; two-tailed Fisher's exact test, p=0.02; p=0.05, respectively; *Figure 5—figure supplement 1c*). We found that skin-enriched alleles were dramatically overrepresented in skin compared to cartilage/bone (14,103-fold greater RPM, s.d.= ±15,680 RPM, n = 14 alleles); thus, these changes in the levels of skin-enriched alleles in the limb likely reflect the migration of epidermal cells from outside of the wound site to contribute to the regenerating skin rather than dermal cells that contribute to cartilage (*Currie et al., 2016*; *Seifert et al., 2012*).

Within the whole hand, such tissue-enriched alleles are almost exclusively found among low frequency alleles (57/58 alleles), and together they represent a minor proportion of the total sequence reads within the hand (mean = 7.8%, s.d. = ±8.6%, n = 4 hands, *Figure 5—figure supplement 1*). Of 58 alleles found to be tissue-enriched, 27 were not detected in sequence reads from the whole hand, and such alleles were exclusively low frequency within the tissues in which they were enriched. Together, all such tissue-enriched alleles that could not be detected within the hand represent 0.43% (s.d. = ±0.63%, n = 24 tissue-enriched samples) of all reads within the tissues in which they are enriched, and thus, were likely under-sampled rather than absent in the hand. These comparisons among alleles from different tissue samples indicate that high-frequency alleles account for most of the sequence reads obtained in single limbs and are broadly distributed throughout tissues. Such alleles either form in the early blastula stages of embryogenesis, or independently in multiple cell lineages, or both. While low-frequency alleles that form at later developmental stages are more likely to exhibit tissue-specific enrichment, such tissue-enriched alleles often occur at such low frequencies as to not be detectable within entire limb samples using NGS.

Despite the relatively minor contribution of alleles identified as tissue-enriched to the entire composition of the hand and limb, we observe that the allelic compositions of different tissue samples vary. We find that the allele identities and frequencies found in nerve, blood vessel, and blood-enriched samples exhibit very low correlations with each other and with those from other tissue-enriched samples (*Figure 5a*). Alleles in the muscle and skin exhibit both higher correlations with each other and with those found within the entire hand than to those of other tissue-enriched samples (*Figure 5a*). These similarities are likely partially due to the overrepresentation of these tissues within the hand, as well as the fact that such dissected samples from the muscle and skin of the limb contain significant contributions from multiple tissues. Nonetheless, as these tissue-enriched samples were derived from fully regenerated limbs, the variation in the allelic compositions of the regenerated tissue-enriched samples indicate that they are differently reconstituted with respect to their embryonic origins.

Our sequencing results indicate that limb regeneration is a high fidelity process as the embryonic lineages that are detected in the original limb are retained after multiple rounds of regeneration at similar frequencies; however, we do find that some alleles present in original limbs were not detected in secondary limbs. This is reflected in the apparent bimodality of the distribution in changes in allele frequencies between primary and secondary limbs depicted in *Figure 2b*. To further investigate whether such alleles are truly lost, we examined whether such alleles were subsequently detectable in limbs from the same animal after additional rounds of regeneration. In the limbs of 4 animals that were examined for three rounds of regeneration, of 54 alleles lost between primary and secondary limbs, 35 were found to be present in subsequent limbs or tissues. While novel alleles appeared in subsequent limbs, 5/5 such alleles detected in the secondary limbs but not

**Figure 4.** Comparison of alleles detected in primary, secondary, and tertiary limbs. (**A**) Schematic depicting quantification of alleles in primary, secondary, and tertiary limbs. Mutagenized primary limbs were amputated, allowed to fully regenerate to produce secondary limbs, again amputated, allowed to fully regenerate to produce tertiary limbs, and amputated. (**B**) Comparison of frequency of all alleles tracked in this study between primary and tertiary limbs of 12 animals. The log scores of the normalized read number of every allele in the original limb are significantly correlated with those of the tertiary limb; data in *Figure 1—source data 1*. (**C**) The total percent changes in allelic identity for each secondary (blue) and tertiary (black) limb

*Figure 4 continued on next page*

*Figure 4 continued*

for each animal when compared to allelic frequencies of previous limb. The asterisk marks the single primary limb that exhibited polydactyly. These data are *Figure 4—source data 1*.

DOI: https://doi.org/10.7554/eLife.25726.011

The following source data and figure supplements are available for figure 4:

**Source data 1.** Mutation frequency and allelic change in all animals.

DOI: https://doi.org/10.7554/eLife.25726.014

**Figure supplement 1.** Spontaneous polydactyly in animal exhibiting dissimilar primary and secondary limb allelic identities.

DOI: https://doi.org/10.7554/eLife.25726.012

**Figure supplement 2.** The log scores of reads per million (RPM) of every allele in the original limb compared with those of the tertiary limb for each individual animal that underwent secondary amputations.

DOI: https://doi.org/10.7554/eLife.25726.013

in the original limbs were found in subsequent regenerated limbs or tissues from the same animal (4/6 alleles detected in the tertiary, but not primary limb, were detected in other iterations of the limb; *Figure 5—figure supplement 2*). The disproportionate number of unrecovered alleles from the primary to secondary limb compared to those from the secondary or tertiary to primary limbs indicates that some cell lineages are truly lost upon limb regeneration while others were not detected, most likely due to under-sampling at the initial DNA amplification step. The average allele present in the original limb that is subsequently undetectable after regeneration is a rare component of the limb (mean = 748 RPM, s.d. =±1656, median = 20 RPM, n = 19/154 alleles detected in the original limb). The sum of such lost cell lineages in each limb investigated represents a minor portion of the cell lineages comprising the limb (0.34%, ±0.56%, n = 4 limbs), indicating that either a subset of embryonically-labeled cell lineages are unable to participate in limb regeneration or that some cell lineages become isolated in distal regions of the limb that are consequently lost after amputation.

In recent years, studies using transgenic tissue grafts in the axolotl have provided a powerful means to track the behaviors of cells and tissues of differing origins during limb regeneration (*Kragl et al., 2009*; *Maden et al., 2015*; *McCusker et al., 2016*; *Nacu et al., 2013*). Such studies have created a growing consensus that the role of transdifferentiation of tissues in limb regeneration is minimal in salamanders, with cells of regenerating tissues largely arising from cells of the same embryonic origin (*Kragl et al., 2009*); yet, the scope of such grafting work has been limited as only the fates of tissues derived from single large tissue grafts could be assessed in one limb at a time. More recent efforts have coupled this approach with tissue-specific transgenic lines and in vivo imaging of cell behaviors during appendage regeneration to elucidate this process with spatial and temporal resolution (*Currie et al., 2016*; *Sandoval-Guzmán et al., 2014*). Such works continue to demonstrate that regenerated tissues arise from tissues of a similar origin, though some ambiguity remains. For instance, two recent studies demonstrate that cartilage and bone make little, if any, contribution to skeletal regeneration (*Currie et al., 2016*; *McCusker et al., 2016*), with dermal fibroblasts appearing to give rise to most regenerative connective tissues during digit tip regeneration (*Currie et al., 2016*). With the genetic labeling approach applied in our study, we can quantify the relative contributions of numerous, often very low frequency cell lineages to limb regeneration at once. We can infer from our data that low frequency populations of cells do not largely repopulate the regenerated limb at the expense of others. Taken together, our data demonstrate that the same embryonic lineages that produce the limb faithfully regenerate the limb in the same proportions. These conclusions do not challenge the recent observations that bone and cartilage give rise to little or no skeletal tissues after regeneration (*Currie et al., 2016*; *McCusker et al., 2016*), but indicate that the cells that replace amputated bone and cartilage are most likely derived from the same embryonic cell lineages that initially formed the lost bone and cartilage.

Here, we report the development of a novel means to characterize the fidelity with which the embryonic cell lineages that give rise to the limb are recapitulated in the regenerated limb, and we have identified several caveats that should be considered in future applications of this approach. Zygotic injection of *cas9* and individual gRNAs achieves a high rate of mutagenesis, but the majority of cells are labeled with mutations that occur in either the earliest cell divisions or multiple times throughout development. Such labeling redundancy could be reduced by the application of multiple

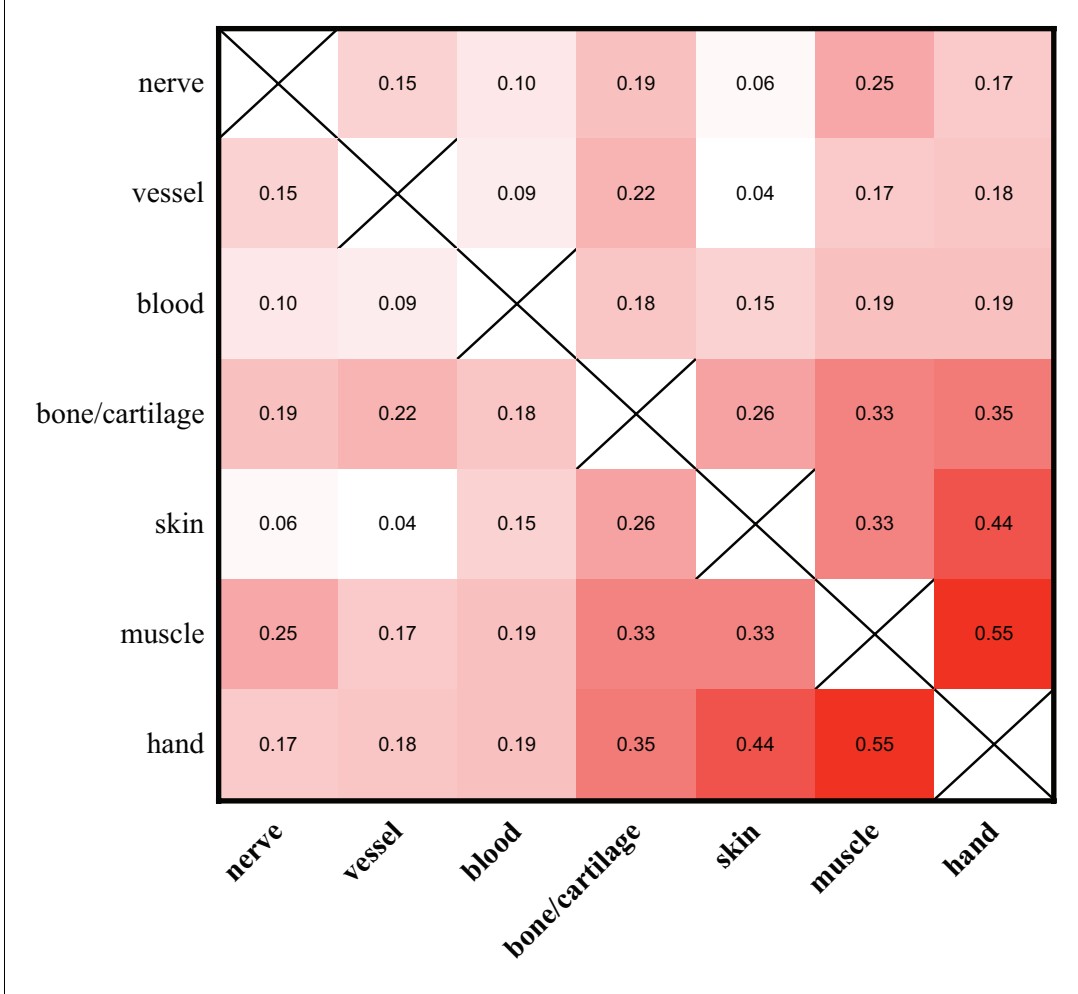

**Figure 5.** Allelic compositions of tissue-enriched samples vary relative to each other. The heat map of the $r^2$ log scores of normalized RPM +1 values for alleles of all tissue-enriched samples from four quaternary limbs and hands. More isolatable samples (nerve, vessel, and blood) display both low correlations to each other and to the hand, while more heterogeneous samples (bone/cartilage, skin, and muscle) are more correlated to each other and to the hand. These data are in *Figure 5—source data 1*.

DOI: https://doi.org/10.7554/eLife.25726.015

The following source data and figure supplements are available for figure 5:

**Source data 1.** All allele counts in all tissues.

DOI: https://doi.org/10.7554/eLife.25726.018

**Figure supplement 1.** Identification and quantification of tissue-enriched alleles.

DOI: https://doi.org/10.7554/eLife.25726.016

**Figure supplement 2.** Alleles are disproportionally lost after first amputation.

DOI: https://doi.org/10.7554/eLife.25726.017

gRNAs within a single amplifiable region. Alternately, while the comparatively long generation time of the axolotl makes the generation of stable, single-copy transgenic lines a time consuming task, synthetic transgenic arrays of CRISPR targets (*McKenna et al., 2016*) may be eventually employed to create higher density allelic maps. While the method described here detects numerous, low-frequency, tissue-enriched alleles, such alleles often occur at frequencies at or beyond the threshold of detection when DNA is sampled from the entire limb. Input DNA quantity and read coverage should be optimized, particularly as the approach is used to characterize the behavior of increasingly low frequency cell lineages.

This study marks the first example by which regeneration has been characterized using a CRISPR-derived lineage tracing approach, and we find that the regenerated limb is repopulated by

embryonic cell lineages in a manner that recapitulates the development of the original limb. As the ability of cell lineages to contribute to a regenerated limb is a quantifiable phenotype that may be associated with mutant genotypes on a large scale within an individual limb using NGS, this approach may be coupled with targeted-mutagenesis of uncharacterized candidate genes to identify novel regulators of limb regeneration. Furthermore, as CRISPR/Cas is widely applicable across species, similar approaches may be used to characterize many regenerative processes across a wide variety of organisms.

## Materials and methods

### Animals

All animal experiments were carried out on *Ambystoma mexicanum* (axolotls) in facilities at Yale University. Experimental procedures were approved by the Yale University IACUC (2016–10557) and were in accordance with all federal policies and guidelines governing the use of vertebrate animals. All axolotls used in this study were produced by natural mating and housed in our facility. They were fed artemia, blood worms, and fish pellets. The parental *cagg:egfp* and *cagg:nuclear mcherry* transgenic animals were originally obtained from the Ambystoma Genetic Stock Center and are previously described (*Kragl et al., 2009*; *Sobkow et al., 2006*). Our previous work has indicated that these transgenic lines contain a single integration of these transgenes (*Flowers et al., 2014*).

### Genome editing

For lineage tracing experiments, we targeted one site in an *egfp* transgene and endogenous *tyrosinase* gene, and two sites in a *mCherry* transgene and endogenous *methyltransferase-like* gene listed in *Supplementary file 1*. Target gRNA sequences are listed in *Supplementary file 1*. Axolotl matings and microinjections were carried out as previously described (*Flowers and Crews, 2015*), using 500 pg of sgRNA and 1000 pg of *cas9* mRNA. *cagg:egfp* and *cagg:nuclear mcherry* transgenic animals were mated with non-transgenic animals. Successful mutagenesis of *egfp* and *nuclear mcherry* transgenic animals was confirmed by imaging animals using fluorescence and a Zeiss stereomicroscope. Successful mutagenesis of *tyrosinase*, *meth* t1, and *meth* t2 were confirmed using fragment analysis of fluorescent PCR products as previously described (*Flowers and Crews, 2015*).

To assess allelic diversity, matings and microinjections were carried out as before but 5 pg of each of 5 sgRNAs and either 100 or 1000 pg of *cas9* mRNA were injected into each blastomere at the two-cell stage. At two weeks post-injection, 5 larvae from each injection condition were used for genomic DNA extraction, target amplification, and NGS.

### Amputations and tissue dissections

Initial amputations were performed when animals were 5-to-6 cm in length (mouth to tip of tail). Secondary amputations were performed 4 months after initial amputations. Tertiary amputations were performed 4 months after secondary amputations. Animals were anesthetized in 1 g/L Tricaine in Holtfreter's solution. All amputation were made through the radius and ulna just distal to the elbow. For each gRNA target, an additional amputation was performed on a non-mutant animal or corresponding transgenic animal to serve as a non-mutant control for sequencing. Limbs were frozen at −20°C until all primary, secondary, and tertiary limbs were collected.

Quaternary limbs were collected from four animals (*egfp 1*, *tyr 1*, *meth t2 2*, and *meth t2 3*) to produce tissue enriched samples 11 months after tertiary amputations. Animals were as anesthetized as before. Full thickness skin was removed from the forelimb after making an incision along the ventral midline of the forelimb from the wrist to the elbow followed by two incisions along the circumference of the forelimb just distal to the elbow and just proximal to the wrist. The flexor muscles of the forearm were displaced with fine tipped forceps to gain access to the underlying radial and ulnar artery and branches of the brachial and ulnar nerves (*Francis, 1934*). The artery was distinguished from the spinal nerves by the presence of circulating blood. The entire length of the artery along the forelimb was collected with forceps followed by the spinal nerves. All tissues distal to the ulna (referred to as the 'hand') were then amputated from the forelimb just proximal to the carpus and collected. The forelimb was fully amputated just distal to the elbow and the muscle was separated from the bone using scalpel and forceps. Blood samples were drawn by inserting the tip of a 28-

gauge insulin syringe into the base of the gills. All tissues were frozen at −20°C until further processing.

## Genomic DNA preparation

Genomic DNA was extracted from entire amputated limbs or larvae using the DNeasy Blood and Tissue Collection Kit (Qiagen). Limbs were digested with vortexing in Proteinase K with for 6 to 8 hr at 56°C until no evidence of solid tissue remained before continuing with DNA extraction. Similarly, tissue-enriched samples and hands were digested 4 to 18 hr as described.

## Target amplification

All PCRs for primary, secondary, and tertiary limbs were performed at the same time. 50 to 100 ng of genomic DNA was used as template for 50 μL reactions using Phusion polymerase. For each target, a standard reverse primer and a unique forward primer containing a unique 4 to 6 bp 5' identifier sequence for each injected animal and non-mutant control (see *Supplementary file 1*) were used. An additional PCR was performed with a unique identifier primer as a no genomic DNA control. All products were purified using QiaQuick PCR Purification Columns (Qiagen).

## Sample processing

All PCR products were quantified using Agilent DNA 1000 Chips on an Agilent Bioanalyzer 2100. The concentration of DNA between 90 and 250 bp for each PCR product was calculated, and 10 ng of each product (and, for negative controls, a volume equivalent to the average volume for each sample) was run through a Blue Pippin 3% agarose gel cassette to size select products within the 90 to 250 bp range. All collected products for each sample were pooled using an equivalent quantity of DNA from each PCR from each respective limb or larva sample.

## NGS

All limb sequencing was performed at the Yale Center for Genome Analysis. Samples were purified using AmpPure XP beads, underwent end-repair, overhang addition, adapter ligation, and qPCR to enrich and validate the library. Paired-end sequencing was performed on an Illumina MiSeq using a 500 bp kit. Tissue-enriched and hand samples were processed as described on a second MiSeq run.

All larval sequencing was performed at the Keck DNA Sequencing Facility at Yale. Template preparation was carried out on a One Touch 2 System, and samples were sequenced using an Ion Torrent Personal Genome Machine on an Ion 318 Chip v2.

## Sequence analysis

Geneious software was used for the identification of CRISPR-generated alleles as it permits the assembly and automated counting of large sets of sequences reads and arranges such reads within an alignment so that reads with similar sequences neighbor each other. All sequences containing perfect sequence matches to each unique sample-identifying primer were assembled into sequence lists. All sequences containing perfect matches to the respective gRNA plus protospacer motif (PAM) sequence were removed from each sequence list, counted, and labelled as non-mutant sequence. Subsequently, the remaining sequences in each sequence list for each limb were aligned to the non-mutant amplicon, permitting gaps of up to 100 bases, to form contigs. Within each contig, sequences were then identified that contained a deletion, insertion, or combination of both that affected at least one base within the PAM sequence or within the target sequence within three bases from the PAM sequence, including large deletions that encompassed both the PAM sequence and entire target sequence. Approximately, 13 to 20 bases on each side of the indel site were incorporated to generate a sequence defined as an allele. All sequences containing this newly defined allele were then removed from the respective contig. This process was repeated for many iterations until the only remaining sequences in each contig were those containing mismatches near to but not immediately flanking the putative gRNA cut site and could thus be regarded as products of PCR and/or sequencing errors. For each respective target, a sequence list was made containing all putative alleles corresponding to all those identified in all animals injected with a given gRNA. Each allele was then aligned to the reference, non-mutant amplicon. A unique name was given to each allele, corresponding to the start and end sites of any deletion with respect to the reference amplicon,

followed by the identity of any insertion. All redundant alleles were eliminated, and all alleles for a respective target were trimmed to encompass the same number of bases (from 27 to 36). This allele list was then aligned so that each allele formed a contig with all sequences from each individual limb that perfectly matched, so that each sequence read could only match with one allele.

Using this protocol, 25,000 to 1,400,000 reads were assigned to each respective limb or tissue-enriched sample. To exclude falsely detected alleles, prior to normalizing the data, all alleles from individual animals were excluded if the raw number of reads for that allele was not at least two-fold greater than the raw number of reads for the same allele detected in the corresponding non-mutant control. Additionally, all alleles detected in individual limbs in which fewer than four reads were detected were excluded. The total number of assigned reads for each limb were then added to obtain the total number of reads for the limb. The total read number was divided by 1,000,000 to get the normalization coefficient, and the raw number of reads for each allele was multiplied by the normalization coefficient to produce Reads per Million (RPM). To obtain coherent log scores for this value, RPM + 1 was used.

Linear regressions were performed using log scores of RPM + 1 for each allele after all alleles in which both log scores were equal to zero were excluded. GraphPad Prism was used to calculate $R^2$ values.

The percent change in allelic identity was calculated as follows for primary and secondary regenerations, respectively:

$$(\sum |\text{RPM allele } a_{primary} - \text{RPM allele } a_{secondary}| + |\text{RPM allele } b_{primary} - \text{RPM allele } b_{secondary}| + \ldots)/2 \times 100\%$$

$$(\sum |\text{RPM allele } a_{secondary} - \text{RPM allele } a_{tertiary}| + |\text{RPM allele } b_{secondary} - \text{RPM allele } b_{tertiary}|\ldots)/2 \times 100\%$$

## Additional information

### Funding

| Funder | Grant reference number | Author |
| --- | --- | --- |
| Connecticut Innovations | Seed Grant 15RMA-YALE-09 | Grant Parker Flowers |
| Eunice Kennedy Shriver National Institute of Child Health and Human Development | Individual Postdoctoral Fellowship F32HD086942 | Grant Parker Flowers |
| Connecticut Innovations | Established Investigator Award 15-RMB-YALE-01 | Craig M Crews |
| National Institute of General Medical Sciences | Predoctoral Training Fellowship T32GM007499 | Lucas D Sanor |

The funders had no role in study design, data collection and interpretation, or the decision to submit the work for publication.

### Author contributions

Grant Parker Flowers, Conceptualization, Data curation, Formal analysis, Funding acquisition, Validation, Investigation, Visualization, Methodology, Writing—original draft, Project administration, Writing—review and editing; Lucas D Sanor, Conceptualization, Data curation, Formal analysis, Validation, Investigation, Writing—review and editing; Craig M Crews, Resources, Supervision, Funding acquisition, Project administration, Writing—review and editing

### Author ORCIDs

Grant Parker Flowers http://orcid.org/0000-0001-7436-3531
Craig M Crews http://orcid.org/0000-0002-8456-2005

### Ethics

Animal experimentation: Experimental procedures were approved by the Yale University IACUC (2016-10557) and were in accordance with all federal policies and guidelines governing the use of vertebrate animals.

Decision letter and Author response
Decision letter https://doi.org/10.7554/eLife.25726.021
Author response https://doi.org/10.7554/eLife.25726.022

## Additional files

**Supplementary files**
• Supplementary file 1. Primers and gRNA target sequences.
DOI: https://doi.org/10.7554/eLife.25726.019

• Transparent reporting form
DOI: https://doi.org/10.7554/eLife.25726.020

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
