## [Decision Letter]

Thank you for submitting your article "Lineage tracing of genome–edited alleles reveals high fidelity axolotl limb regeneration" for consideration by *eLife*. Your article has been reviewed by three peer reviewers, one of whom, Alejandro Sánchez Alvarado is a member of our Board of Reviewing Editors, and the evaluation has been overseen by the Reviewing Editor and Patricia Wittkopp as the Senior Editor. The following individual involved in review of your submission have agreed to reveal their identity: James Robert Monaghan (Reviewer #3).

The reviewers have discussed the reviews with one another and the Reviewing Editor has drafted this decision to help you prepare a revised submission. We hope you will be able to submit the revised version within two months.

Summary:

Whether regenerated limbs arise from dedifferentiation, proliferation of stem cells or transdifferentiation of existing tissue types is an important question. The authors have established a powerful approach to address this question. This study used CRISPR/Cas9 to create mutations in genes and then next generation sequencing to characterize the frequency distribution of alleles distributed among limb cells before and after amputation. This study is interesting because CRISPR/Cas9 creates highly mosaic animals that contain many unique, moderate–to–low frequency alleles. This allows temporal resolution of many different allelic types during the regeneration process. The authors find that allele frequencies did not overtly change and thus the results suggest that regenerated limbs are similar in cellular composition to the pre–amputated, intact limb. However, no unambiguous answers are provided in the current manuscript. Cell lineages contributing to the regenerated limbs could be from differentiated cells, or progenitor cells. Hence, the central question remains largely unaddressed. Nevertheless, the work is an important piece of information that supports traditional grafting techniques that had suggested similar conclusions.

Essential revisions:

While the fidelity of allele frequencies maintained during regeneration is very exciting, the implications of this finding need further discussion or experimentation. There is little reason to expect that cellular identities and proportions will not be faithfully reproduced after regeneration, assuming, of course, that the effects of metamorphosis or aging were not to play a central role in setting cellular identity. Thus, it's not surprising to see the allele frequencies reproduced, with the results verifying that multiple early embryonic lineages contribute to the development of the limb, and that such lineages contribute in similar frequencies to regenerated limbs. As the limb is a complex system with multiple tissues and cell types, how these lineages translate to tissue or cell types is unclear. It is likely that these alleles mark all necessary progenitors of the cell types in a limb. The fidelity reflects the cell type reconstitution in the new limb. As the number of lesions is not exhaustive, other possibilities remain equally likely. For instance, the alleles may mark some differentiated cell types which will dedifferentiate, or only several types of stem/differentiated cells that matter for the regeneration of that particular group of cells. Hence, this manuscript would be strengthened if experimental data or at least discussions are provided for the following questions.

1) Establish a link between the cells and alleles. Experimentally, the authors can examine the allele frequencies in the blastema and proximal tissues post amputation. The proximal tissues can be dissected into multiple tissue types (e.g., dermis, cartilage, tendons, nerves, and blood vessels). Allele identities and frequencies can be examined in these different tissue types to establish if some alleles can specifically mark tissue types. More rigorously, such analyses can be performed in isolated differentiated cells and proliferating cells of different tissue types. This should be able to address the central question of whether dedifferentiation, transdifferentiation or proliferation of stem cells are involved in the regeneration of the limbs. Furthermore, allele frequencies in the proximal tissues can be compared to old and new limbs at the tissue or cell type level to investigate this question.

2) Figure 3: Discuss the reasons for the emergence of new alleles and loss of previous alleles in regenerated limbs (e.g., cell behaviors).

3) What the approach doesn't provide is spatial resolution of alleles, which would be interesting to know so as to answer some of the questions posed in the introduction, for example do cells switch identities during regeneration. What the results suggest is that CRISPR/Cas9 creates the same allele frequency distribution within tissue–specific, progenitor cell populations, whether these populations derive from fated or stem–like cells. However, more is actually known about the relative contribution of differentiated and stem–like cells to limb regeneration and the introduction needs a bit more balance.

4) The article could benefit from a bit more consistency in stating what can be learned using the approach, what was learned, and maybe also mention caveats in the final paragraph along with prospects for future studies. For example, the following sentences are qualitatively different:

"8) We found that regenerated limbs are often nearly identical to the originals even after repeated amputations."

"38) We find that a regenerated axolotl limb is a high fidelity replicate of the original limb".

5) There seems to be a lack of discussion in the findings of this study in the context of what is known about axolotl limb regeneration from previous literature. A key discussion point that needs to be included is to address Currie et al., 2016. This is a critical point of comparison that needs to be discussed before publication.

6) Figure 1: Why are there so many alleles that completely lost in the regenerated limb and vice versa? This suggests that certain cell types may be behaving differently. There is evidence for old muscle not generating muscle in regenerated limbs, instead by satellite cells (Sandoval–Guzman et al., 2014). Are the alleles lost in the regenerating limbs representing the muscle fibers that do not contribute to the regenerated limb? Also, do alleles that are found only in the regenerating limb arise from distant cells such as blood cells or Schwann cells? All of these problems cannot be determined by the current approach, but should be discussed.

7) Results and Discussion: There seems to be a bimodal distribution in alleles gained or lost in Figure 2. This would again suggest that certain populations of cells are lost from the regenerate and others are provided from a distant source or from a small population of cells in the uninjured limb. So yes, the conclusions that there are a significant number of cells that do not change in frequency is true, it is unclear what types of cells these are.

8) Results and Discussion: I partially agree with this statement. The results of McKenna et al., 2016 suggest that adult tissues in zebrafish are derived from few embryonic progenitors. It is possible that high similarity may be partially due to alleles represented in all the cells of a particular cell type, like muscle (or other cell type) that came from the same few progenitors. Having a general indication for what cell types are carrying specific alleles would be very helpful in interpreting the data.

---

## [Author Response]

Essential revisions:1) Establish a link between the cells and alleles. Experimentally, the authors can examine the allele frequencies in the blastema and proximal tissues post amputation. The proximal tissues can be dissected into multiple tissue types (e.g., dermis, cartilage, tendons, nerves, and blood vessels). Allele identities and frequencies can be examined in these different tissue types to establish if some alleles can specifically mark tissue types. More rigorously, such analyses can be performed in isolated differentiated cells and proliferating cells of different tissue types. This should be able to address the central question of whether dedifferentiation, transdifferentiation or proliferation of stem cells are involved in the regeneration of the limbs. Furthermore, allele frequencies in the proximal tissues can be compared to old and new limbs at the tissue or cell type level to investigate this question.

We carried an experiment very similar to that suggested by the reviewers. We have multiple animals with well–characterized mutant allele populations. In these animals, we carried out amputations and dissections from the quaternary limbs. There were several challenges to connecting cell types to alleles that we had to take into account while designing our approach. First, if a particular allele were found in an isolated population of cells, all other cell types would have to be parsed before determining that the allele is cell specific. Second, as full limb regeneration takes several months, we set out to collect data from animals for which an allelic data set already existed. Thus, we carried out dissections from the large, fully regenerated limbs of animals used in the experiments described previously in the paper. This allowed us to include almost all the tissue comprising the regenerated limbs in our tissue samples. As described in the methods, we were able to dissect the radial and ulnar arteries, as well as the brachial and ulnar nerves in the forearm proximal to the carpus. From the same animals, we collected full thickness skin and removed the muscle surrounding the radius and ulna. We collected the entire radius and ulna, all the remaining distal tissue (the “hand”) for comparison, and blood from the gills of the animal.

We expected to see considerable overlap in alleles between these tissue–enriched samples due to the heterogeneity of cell types and the shared embryonic origins. Thus, we defined alleles as tissue–enriched if one or two tissues were found to exhibit 16–fold more normalized reads compared to all others. While these criteria led us to identify numerous alleles that were tissue–enriched, we found that many of their frequencies were below the threshold of detection within the limb and hand samples. This finding suggests that, while tissue–specific alleles likely exist that uniquely identify various types of differentiated cells, they are often undetectable when sampling from the entire limb. This data is visually represented in Figure 5—figure supplement 1.

Nonetheless, we find that tissue samples differ in their allelic compositions. Tissues tend to be more similar to the hand than they are to one another and samples that are more heterogeneous are more similar to each other (Figure 5). This approach allowed us to retrospectively analyze data from our previous limb experiments and make insights into the tissues where specific alleles were found.

A NGS approach of CRISPR–induced alleles could be a powerful tool for analyzing the origins and relative contribution of transdifferentiation, dedifferentiation, and stem cells to specific tissue types in regenerating limbs; however, this requires surmounting a number of technical limitations, which we discuss. We are currently pursuing approaches that allowed for more detailed analyses.

2) Figure 3: Discuss the reasons for the emergence of new alleles and loss of previous alleles in regenerated limbs (e.g., cell behaviors).

Analyzing a fourth limb, for our tissue analysis, allowed us to better characterize alleles from our previous experiments. We found that many alleles in the original limb are not detected in the secondary limb. However, most of these alleles reappear in tertiary limbs or in the tissues from the quaternary limb, indicating that they are not truly lost. In some cases new alleles appeared in the secondary limb. These were located in dissected tissue samples and were found to be enriched in the muscle and blood; however, these findings were not statistically significant. Our data indicate that many alleles that appear to be lost in the first limb or are gained in the secondary, are actually an artifact of under–sampling. Since the primary limb is the smallest, it contains the fewest cells of all limbs analyzed. Therefore, a large selection of alleles will be represented within the sampled DNA. As the limb grows, there will be larger populations of cells with similar mutations that dilute the diversity of the cells and alleles sampled. Though true loss of alleles likely also contributes, undersampling of DNA as the animals increase in size contributes to the disproportionate loss of alleles from the first limb to the second. We found that unrecovered alleles represent less than 1% of the original amputated tissue, and speculate that these may have been cell lineages restricted to distal amputated portions of the limb.

Our data on tissue–enriched alleles allowed us to assess whether cells of a particular lineage were more likely to change than others. We found no significant differences between tissues in the number of alleles that increased or decreased in frequency from one limb to the next, with the exception of skin–enriched alleles, which disproportionately decreased from the primary to secondary and disproportionately increased from the secondary to tertiary. The sample sizes are comparatively small, allowing for the chance occurrence of such a low P value seen after 36 comparisons. We speculate that these changes may arise from the migration of highly motile epidermal cells after amputation.

3) What the approach doesn't provide is spatial resolution of alleles, which would be interesting to know so as to answer some of the questions posed in the introduction, for example do cells switch identities during regeneration. What the results suggest is that CRISPR/Cas9 creates the same allele frequency distribution within tissue–specific, progenitor cell populations, whether these populations derive from fated or stem–like cells. However, more is actually known about the relative contribution of differentiated and stem–like cells to limb regeneration and the introduction needs a bit more balance.

We very much agree with this statement and acknowledge that allele frequency distributions of differentiated cells are very likely the same as those found in their progenitors. We have modified the text to clarify this and we have commented on complexity of parsing the contributions of different cell types in our introduction and discussion.

4) The article could benefit from a bit more consistency in stating what can be learned using the approach, what was learned, and maybe also mention caveats in the final paragraph along with prospects for future studies. For example, the following sentences are qualitatively different:"8) We found that regenerated limbs are often nearly identical to the originals even after repeated amputations.""38) We find that a regenerated axolotl limb is a high fidelity replicate of the original limb".

We agree and have modified the discussion to more consistently describe the conclusions that can be made using our approach. We have eliminated the statement that “regenerated limbs are often nearly identical.” Instead we emphasize that our study provides an important information about the baseline level at which alleles are retained in regenerating limbs. This provides an important framework for the future studies using mosaic gene–editing with NGS to determine the fidelity of regeneration. Instead, we have expanded our discussion of caveats, particularly with regard to the timing of mutagenesis and sampling of DNA.

5) There seems to be a lack of discussion in the findings of this study in the context of what is known about axolotl limb regeneration from previous literature. A key discussion point that needs to be included is to address Currie et al., 2016. This is a critical point of comparison that needs to be discussed before publication.

We have improved our discussion of previous literature, including mention of Currie et al. We believe that our findings are consistent with Currie et al., as the dermal fibroblasts and the chondrocytes they replace likely share the same embryonic lineage.

6) Figure 1: Why are there so many alleles that completely lost in the regenerated limb and vice versa? This suggests that certain cell types may be behaving differently. There is evidence for old muscle not generating muscle in regenerated limbs, instead by satellite cells (Sandoval–Guzman et al., 2014). Are the alleles lost in the regenerating limbs representing the muscle fibers that do not contribute to the regenerated limb? Also, do alleles that are found only in the regenerating limb arise from distant cells such as blood cells or Schwann cells? All of these problems cannot be determined by the current approach, but should be discussed.

We agree that the identity of lost alleles is of great interest, but we cannot devise any way to address to retroactively identify lost alleles. Our investigation of several regenerated limbs and tissue–enriched samples suggest that many lost alleles have not been truly lost, but only appeared lost due to under–sampling. A portion of alleles in the first limb, that make a minor contribution to the total limb, do appear to be permanently lost, and this is now discussed in the text.

According to Sandoval–Guzman et al., muscle fibers in regenerated limbs arise from satellite cells. This suggests that, in regenerated limbs, satellite cells and muscle fibers will be labeled with the same alleles. Since our tissue–enriched allele studies were performed on regenerated muscle, we were therefore unable to pursue the possibility that the lost alleles are present in muscle fibers..

Curiously, our nerve–derived tissue samples (which most likely consisted of DNA derived from Schwann cells and connective tissue) contained only one allele that could be characterized as nerve–enriched. The frequency of this allele remained constant throughout earlier iterations of the limb, indicating that we are unable to detect the migration of new Schwann cells.

As discussed above, we found that new alleles that arose in regenerated limbs were enriched in muscle and blood, but this was not statistically significant. We have added extensive discussion regarding the loss and gain of alleles in the regenerated limb.

7) Results and Discussion: There seems to be a bimodal distribution in alleles gained or lost in Figure 2. This would again suggest that certain populations of cells are lost from the regenerate and others are provided from a distant source or from a small population of cells in the uninjured limb. So yes, the conclusions that there are a significant number of cells that do not change in frequency is true, it is unclear what types of cells these are.

We have added discussion about the bimodal distribution of alleles into the text. This apparent bimodality, particularly for the low frequency alleles, partially arises from an under–sampling of cells. We can say this because many of these alleles were detected again in subsequent limbs or specific tissues. Sampling errors likely account for a portion of the alleles present at very low frequencies in the original limbs and exhibit several fold frequency increases in subsequent limbs. A subset of alleles that increased in frequency and were retained after regeneration can, however, be assigned to individual tissues. These alleles were distributed among bone, blood, and muscle, though these overrepresentations are not statistically significant.

We agree that the identity of the cell lineages that expand is fascinating; however, such individual lineages represent less than one–thousandth of the limbs, making it very challenging to identify their cellular identity. We are actively exploring new methods that may allow us to identify these types of cell lineages.

8) Results and Discussion: I partially agree with this statement. The results of McKenna et al., 2016 suggest that adult tissues in zebrafish are derived from few embryonic progenitors. It is possible that high similarity may be partially due to alleles represented in all the cells of a particular cell type, like muscle (or other cell type) that came from the same few progenitors. Having a general indication for what cell types are carrying specific alleles would be very helpful in interpreting the data.

As discussed previously, we have devoted considerable effort toward isolating and sequencing from tissues to identify specific allele–labeled cell types. To conclusively assign an allele to a unique cell type requires sampling from the entire limb rather than from isolated cells. Our data indicates that while many alleles are shared between several tissue types, a subset of alleles (representing slightly less than one third of those detected) were enriched more than 16–fold in one or two tissue samples. These alleles were almost exclusively very low frequency, and approximately half were not detected within the adjoining hand. Thus, while cell–type–specific alleles exist, they are both difficult to isolate, and occur at such low frequencies that it is difficult to reliably trace them in the tissue of entire limbs, especially in larger animals. Nonetheless, we find that tissues, especially those that are easier to isolate, have distinct allelic composition patterns, reflecting different relative contributions of embryonic lineages.